# Patterned crystal growth and heat wave generation in hydrogels

Thomas B. H. Schroeder [1✉] & Joanna Aizenberg [1,2✉]

The crystallization of metastable liquid phase change materials releases stored energy as latent heat upon nucleation and may therefore provide a triggerable means of activating downstream processes that respond to changes in temperature. In this work, we describe a strategy for controlling the fast, exothermic crystallization of sodium acetate from a metastable aqueous solution into trihydrate crystals within a polyacrylamide hydrogel whose polymerization state has been patterned using photomasks. A comprehensive experimental study of crystal shapes, crystal growth front velocities and evolving thermal profiles showed that rapid growth of long needle-like crystals through unpolymerized solutions produced peak temperatures of up to 45 ˚C, while slower-crystallizing polymerized solutions produced polycrystalline composites and peaked at 30 ˚C due to lower rates of heat release relative to dissipation in these regions. This temperature difference in the propagating heat waves, which we describe using a proposed analytical model, enables the use of this strategy to selectively activate thermoresponsive processes in predefined areas.

[1] John A. Paulson School of Engineering and Applied Sciences, Harvard University, Cambridge, MA, USA. [2] Department of Chemistry and Chemical Biology, Harvard University, Cambridge, MA, USA. ✉email: tschroeder@g.harvard.edu; jaiz@seas.harvard.edu

Thermally responsive materials such as hydrogels, liquid crystals, shape-memory alloys, and waxes display temperature-dependent behaviors that can transduce heat into optical, mechanical, or chemical changes. Such materials have found wide applications, including in cell culture and tissue engineering[1,2], antifouling surfaces[3], microfluidics[4,5], soft robotics[6,7], and as a component of complex cascades[8,9]. Most deployments of thermally responsive materials rely on bulk changes in temperature to produce a spatially homogeneous response; some produce more complex responses by judicious arrangement of the responsive domains of the material[6]. Coupling such materials to an upstream heat source that dynamically produces thermal patterns with spatial resolution is another way to impart complexity to their responses. This is frequently accomplished using photothermal[2,10] or ohmic[11] heating. However, these methods have limitations. Photothermal heating requires transparent samples, a suitably high-powered light source, and a means of patterning light and moving this pattern with sufficient resolution at the point of use. Achieving patterned ohmic heating requires a connection to external power and complex electrical circuitry—individually addressable resistive heating elements must be distributed throughout the area of interest, and a control scheme must be used to activate them sequentially. The development of materials with dynamic surface topography is a focus of our group; we have applied such surfaces to antifouling applications[12], the movement of cargo[3], the control of cultured cells[2], and the actuation of rigid structures[13]. We are therefore interested in developing heat sources that store both the energy and the pattern required to induce complex frontal behaviors such as patterned waves in thermally responsive materials on demand. For example, a scheme using dynamic swelling or contraction waves to move cargo on a gel film toward a destination requires the swelling or contraction to be directional[14]; such directionality must be programmed. Likewise, rigid gel-embedded structures bend in a direction defined by the gradients involved in the gel's contraction[15]; controlling the direction of actuation requires programming the gradients. Predefining such patterns using a scheme that also stores the energy required to operate may enable such dynamic surfaces to be incorporated into devices that can be used in the field.

When a compound melts, it consumes a quantity of energy known as the enthalpy of fusion or latent heat ($\Delta H_{fus}$) while holding steady at its melting temperature. Conversely, crystallization is accompanied by the release of $\Delta H_{fus}$, generally before the temperature drops below the melting point. The robustness and reversibility of this effect has led to the widespread adoption of phase change materials to store energy, recover energy from waste heat, and buffer against temperature fluctuations. Phase change materials are used for passive thermal management in the context of batteries and electronic devices[16,17], clothing[18], and buildings[19]. Some phase change materials such as salt hydrates can be cooled far below their melting temperatures without freezing, reaching a metastable supercooled state that can rapidly solidify and release energy upon the introduction or formation of a stable crystal nucleus[20]. Supercooled phase change materials have found applications as triggerable heat sources in the context of portable hand heating[21], cold-start automotive engine heating[22], building-scale air and water heating[23], and long-term solar energy storage[24]. These applications are mostly intended to raise temperatures in a bulk volume for human comfort or improved device function; accordingly, the spatial and temporal evolution of the thermal profiles produced by triggerable phase change materials have rarely been engineered beyond the shape of the reservoir. However, as the solidification of metastable phase change materials generally proceeds from a defined nucleation point, the ability to control the velocity of the crystallization front

throughout a reservoir may be useful, enabling control over the time-course of heat release to produce delays or pulses as desired.

Here we show a strategy for patterning crystal growth in metastable supersaturated salt solutions, enabling the creation of spatially programmed dynamic thermal fronts. In order to exert local control over crystallization and heat release, we have incorporated metastable aqueous salt solutions into hydrogels, which can serve as conveniently moldable reservoirs for electrolyte solutions while also being able to regulate transport processes via molecular-scale ion–polymer interactions (Fig. 1a). Previous studies of hydrogels containing metastable solutions[25–31] have focused on the changes in mechanical and transport properties that result from crystallization rather than examining crystal growth kinetics, spatial patterning, or heat evolution in the polymer environment in detail. We demonstrate that polymerizable additives inhibit crystal growth to a significantly greater degree in their polymerized form than as monomers, a finding corroborated by recent work by Deng et al.[32]. This phenomenon enables the patterning of pathways for rapid crystal growth by employing UV-initiated polymerization through photomasks. Upon nucleation with a seed crystal of the salt hydrate, growth proceeds rapidly through the masked areas containing unpolymerized material before slowly proliferating through the bulk. During crystal growth, $\Delta H_{fus}$ is released predominantly at the crystallization front and subsequently dissipates by diffusive heat transfer into the surrounding environment, leading to wavelike temperature profiles. This heat can be harnessed to initiate downstream processes, transducing the initial nucleation stimulus into spatially and temporally programmed responses.

## Results and discussion

**Crystal growth experiments**. We selected sodium acetate trihydrate as the phase change material for this study because it is well-characterized[23], can be stably supercooled to room temperature, crystallizes quickly upon nucleation, has a reasonably high melting point and a high enthalpy of fusion for its material class[20], and can exist stably in supersaturated aqueous solutions for periods on the order of months before crystallizing[20,33]. We prepared metastable precursor solutions containing various proportions of acrylamide monomer, $N,N'$-methylenebisacrylamide as a crosslinker, α-ketoglutaric acid as a photoinitiator, and 7.0 M sodium acetate in water by mixing the ingredients together in a vial using an orbital shaker at 65 °C for at least one hour, until all the solids had dissolved (Supplementary Table 1). We then injected these metastable solutions into flat channels composed of a spacer sandwiched between a glass slide and a coverslip, each functionalized using (3-trimethoxysilyl) propyl methacrylate to prevent the formation of a depletion layer in polymerized solutions[34] and mitigate dewetting induced by the densification of the solution during solidification. After applying a photomask directly onto the coverslip, we cured each solution for 10 min under ultraviolet light, yielding hydrogels or polyacrylamide solutions with a visible phase boundary at the interface between the masked and unmasked areas. Introducing a seed crystal of sodium acetate trihydrate at the edge of the solution induced crystal growth, which proceeded quickly through the masked regions and more slowly through the unmasked regions, allowing spatial control over the time-course of crystallization (Fig. 1a, b, Supplementary Movie 1). The crystals that grew in the masked regions were long and needle-like, whereas the crystals that grew in the unmasked regions were far smaller and more irregular in shape (Fig. 1c, Supplementary Figs. 1, 2, Supplementary Movies 2 and 3). Secondary nucleation of new crystals ensured that the spread of the crystallized region was approximately isotropic in both the masked and unmasked regions, particularly in the unmasked region. Crystals often grew quickly along the interface between the solution and air or the spacer, likely

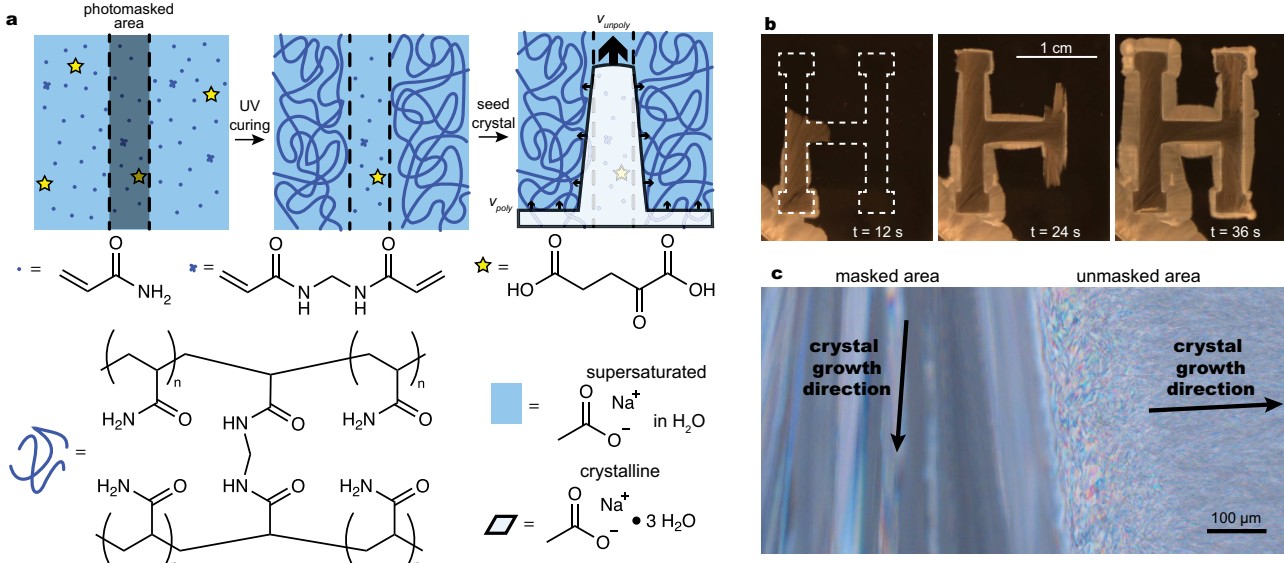

**Fig. 1 Strategy for patterning crystal growth using photomasks and polymerizable additives in solution. a** Scheme showing sequence of events. First, an aqueous prepolymer solution containing acrylamide monomers, *N,N'*-methylenebisacrylamide (a crosslinker), α-ketoglutaric acid (a photoinitiator), and supersaturated sodium acetate is injected into a flat channel and overlaid with a photomask. Next, the system is exposed to ultraviolet light, causing photopolymerization to selectively occur in unmasked areas. Finally, the system is nucleated with a seed crystal of sodium acetate trihydrate, resulting in fast crystal growth through the masked, unpolymerized areas (velocity shown as $v_{unpoly}$) and slower crystal growth through the unmasked regions (velocity shown as $v_{poly}$, where $v_{poly} < v_{unpoly}$). **b** Photos taken at 12-s intervals of sodium acetate trihydrate crystal growth through and around a masked region shaped like the letter H. Precursor solution composition: 7.0 M sodium acetate, 2.8 M acrylamide, 26 mM *N,N'*-methylenebisacrylamide, 2 mM α-ketoglutaric acid in water. Full video available (Supplementary Movie 1). **c** Photomicrograph of transmitted light through a crystalline sample between crossed linear polarizers. This image shows the interface between the masked and unmasked regions of a sample after crystallization; the birefringence of the crystals enables the visualization of discrete crystalline domains, which are large in the masked region and small in the unmasked region. Videos capturing the temporal evolution of crystal growth using this technique are available (Supplementary Movies 2 and 3); electron micrographs of the unmasked region at greater magnification are available in Supplementary Figs. 1 and 2. Precursor solution is the same as in (**b**).

due to the presence of an unpolymerized interfacial layer from oxygen inhibition of the radical polymerization reaction.

In order to establish trends in crystal growth rates with respect to the composition of the solutions, we recorded videos of the propagation of the crystallization fronts through acrylamide solutions of various concentrations in both the polymerized and unpolymerized states and monitored the linear crystal growth front velocity using tracking software. Crystal growth fronts moved at a constant velocity over time except in situations where fronts converged. Such situations result in sufficient warming of the region to enter a temperature regime in which crystal growth kinetics is limited by heat diffusion rather than interfacial attachment[35,36], resulting in decreasing velocities as the region between the converging fronts accumulates heat. To suppress such effects and obtain well-defined readings of the crystal growth rates, we placed our crystallizing samples on a Peltier cooling plate held at 20 °C and recorded growth front velocities in nonpatterned samples (Fig. 2a, b).

**Growth rate trends and possible mechanisms.** Increasing the concentration of acrylamide monomers within the precursor solutions significantly slowed crystal growth in both polymerized and unpolymerized solutions. Crystal growth in polymerized solutions was consistently slower than in unpolymerized solutions; the factor by which polymerization suppressed the growth rate increased with increasing monomer concentration (Fig. 2a). Interestingly, the crystal growth rate in polymerized solutions was not significantly changed by the addition of a cross-linker at concentrations in the range of 0–20 mM (Fig. 2b); however, we included one in photomasking experiments in order to prevent the loss of the pattern via diffusion over time.

There are several apparent possible mechanisms by which polymerized additives might act as stronger crystal growth inhibitors than monomers do; these are discussed in depth in Supplementary Information Section 3, but a brief summary is included here. First, acrylamide monomers' inhibitory activity is well-characterized by several models in which additives adsorb to the surface of the advancing crystal, preventing the further progression of steps along the surface terrace; the extent of inhibition is related to additive concentration via a binding isotherm (Supplementary Fig. 4)[36–41]. The parameters that contribute to inhibition in these models include the additive's binding affinity with the crystal surface and an effectiveness factor that includes steric effects. It is plausible that one or both of these parameters would increase upon polymerization: the steric bulk of a polymer is significantly higher than that of a monomer, and the binding affinity of polymers containing multiple groups that interact with the crystal surface have been shown to bind more strongly and inhibit crystal growth far more effectively[41,42]. Another possible mechanism involves the thermodynamics of crystal growth through porous media[43–46]. Even un-cross-linked polymers in semi-dilute solution form a mesh structure which is likely to be approximately static compared to the high crystal growth velocities considered in this work; the characteristic mesh size decreases with increasing polymer concentration (Supplementary Figs. 2 and 3, Supplementary Table 2)[47]. To grow through small pores, crystals must adopt narrow geometries with highly curved interfaces; the Laplace pressure associated with this curvature depresses the melting point of the crystallizing material and thus the driving force for crystallization, lowering the growth velocity[43,44,48]. A third framing invokes the fact that sodium acetate trihydrate crystals grow in thin needles in which the active growth face has a small area. Upon

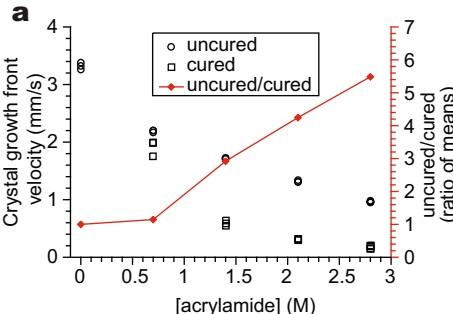 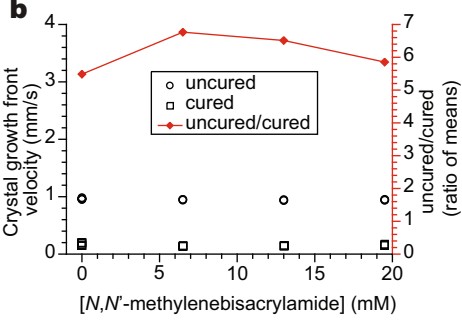

**Fig. 2 Comparison of linear growth rates of sodium acetate trihydrate crystals in samples containing 7.0 M sodium acetate and various polymerized and unpolymerized additive compositions.** $N = 3$ for all points except 2.8 M acrylamide without cross-linker in both graphs, where $N = 6$. Crystallizing samples were placed on a cold plate held at 20 °C. Uncured/cured ratios are ratios of mean velocities. **a** Comparison of growth rates through polymerized and unpolymerized solutions containing different concentrations of acrylamide without a cross-linker. **b** Comparison of growth rates through polymerized and unpolymerized solutions containing 2.8 M acrylamide and various concentrations of $N,N'$-methylenebisacrylamide cross-linker.

contact with the polymer mesh, mass transport resistance to this active face may increase sharply[36,49], effectively halting growth; the growth front speed would then be limited by the kinetics of secondary nucleation. Simulations of this scenario indicate that increased polymer volume fractions lead to slower growth front velocities in a manner that depends on the nucleation rate (Supplementary Fig. 5). A final possible explanation is simply that polymer solutions impede the mass transfer processes involved with crystal growth[26,50]. Each of these effects may play some role in suppressing the velocity of the crystallization front in polymerized systems.

**Wave-like temperature profiles and heat maps.** As we aim to leverage the ability to pattern the evolution of exothermic crystal growth in order to create dynamic heat maps for downstream transduction, we next sought to characterize heat transfer in the crystallizing systems. We recorded videos of crystal growth in solutions and hydrogels using an infrared camera, which provided temperature maps of the system with spatial and temporal resolution (Figs. 3 and 4, Supplementary Movies 4 and 5). When crystals grew through nonpatterned solutions on a cold plate, the temperature profile along the direction of crystal growth quickly reached a steady state relative to the position of the growth front in both the unpolymerized (Fig. 3a) and polymerized (Fig. 3b) cases. The peak temperature in an unpolymerized sample (45 °C) was 15 °C higher than in a polymerized sample (30 °C).

We derived a one-dimensional analytical model to describe these profiles (Supplementary Information Section 4, Eqs. (18) and (19)), revealing that this difference in peak temperature is fully attributable to the difference in crystal growth front velocity (and thus latent heat release) between the two cases. Critically, in both theory and experiments, the profiles only reach a steady state when the system is abutted by a good heat conductor (compare Fig. 3a, b to Supplementary Fig. 6). Accordingly, in subsequent demonstrations requiring large, well-defined temperature differences between masked and unmasked areas, crystallization channels were placed on a metal heat sink. Globally fitting the temperature profiles of unpolymerized and polymerized solutions using shared parameters and the observed velocities yielded reasonable parameter values (Supplementary Table 6) and curves that generally agreed with the observed profiles (Fig. 3a–c). The model overestimates the temperature peak at the crystallization front in unpolymerized samples; this discrepancy may be attributed to the roughness of the advancing front and the dampening of the steep gradient at this point within the cover glass between the solution and the camera. A secondary peak visible at later times in the unpolymerized sample (Fig. 3a)

corresponds to a secondary nucleation cascade that occurs regularly in unpolymerized regions leading to bright whiteness; this is also visible in Fig. 4a and Supplementary Movies 2–4. This cascade seems to consist of small crystallites growing within the interstitial spaces left between the large needles that initially occupy the solution (Supplementary Movies 2 and 3); the small rise in temperature relative to the primary peak reflects the comparatively small volume of these interstices.

The large discrepancy in peak temperatures between polymerized and unpolymerized regions has allowed us to use the photomasking strategy to divide samples into hot regions that undergo fast crystallization and cooler regions in which crystallization is slower, forming complex custom dynamic heat maps (Fig. 4a, b, Supplementary Movie 4). In practice, the boundary on a temperature profile between masked and unmasked regions is not sharp; as the masked regions crystallize, some heat migrates to neighboring areas by lateral diffusion. However, these materials are intended as heat sources to activate downstream thermoresponsive processes. As some processes, such as phase changes, proceed at defined threshold temperatures, it is physically meaningful to discretize these temperature landscapes into regions that reach a threshold temperature and regions that do not (Fig. 4b), as the resulting binary pictures resemble maps of the areas in which downstream processes would be allowed to proceed. To prevent diffusion from obscuring the definition of these maps, the photomasks should be sufficiently larger than the length scale of diffusive heat transfer in the system; accordingly, using thick spacers and wide mask lines helps to retain high temperatures within the desired regions. Additionally, unmasked areas in which fronts converge can reach high temperatures; crystallizing the samples on a heat sink helps to mitigate this, though not entirely (see high-temperature edge regions and interior of "O" and second "L" in Fig. 4b). By using an insulating environment rather than a heat sink, this phenomenon can be intentionally exploited to controllably generate the warmest regions in the sample by using masked areas to define regions where fronts will converge, particularly if mask lines are kept thin (Fig. 4c, d, Supplementary Movie 5).

**Patterning downstream thermoresponsive processes.** Finally, the temperature profiles produced by crystal growth in patterned hydrogels can be coupled to downstream thermally activated effects such as phase transitions in polymers or wax. To demonstrate this, we overlaid a thick (2 mm), fully polymerized crystallization channel with a thin glass coverslip bearing a 160 μm layer of hydrated poly($N$-isopropylacrylamide). This polymer is well-known for forming hydrogels that undergo a

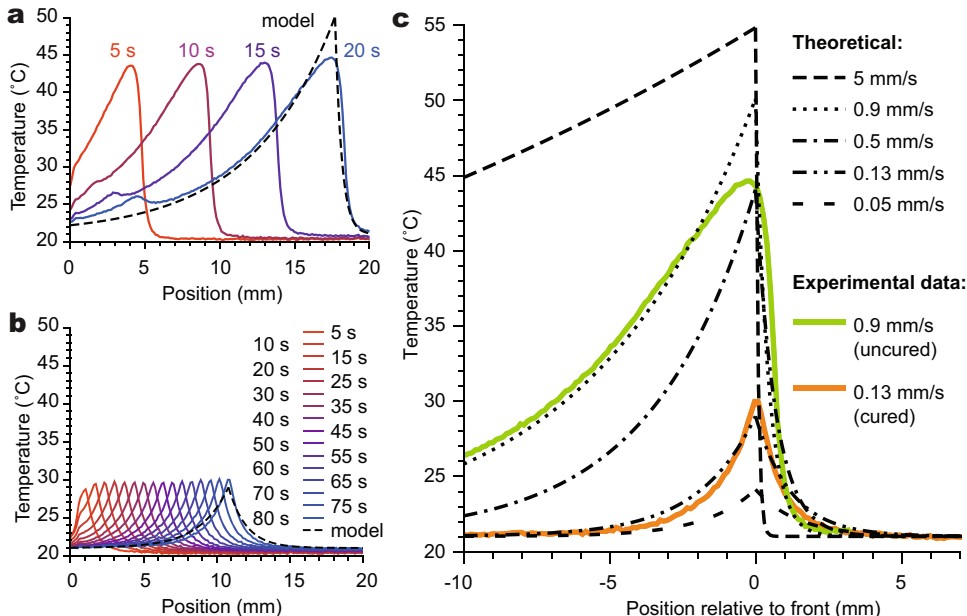

**Fig. 3 Thermal profiles from sodium acetate trihydrate crystal growth through unpolymerized and polymerized samples.** Temperature was monitored with an infrared camera. Solutions were crystallized on a cold plate set to a constant temperature in order to promote heat dissipation. Precursor solution composition for all samples: 7.0 M sodium acetate, 2.8 M acrylamide, 13 mM $N,N'$-methylenebisacrylamide, 2 mM α-ketoglutaric acid in water; 0.5 mm spacer. **a** and **b** Spatial temperature profiles at 5-s intervals resulting from growth of sodium acetate trihydrate crystals through aqueous solutions of unpolymerized (**a**) and polymerized (**b**) additives on a cold plate set to 20 °C. Dashed black lines show results of a global fit of the final curves in both graphs to an analytical model derived in Supplementary Information Section 4. See Supplementary Fig. 6 for similar profiles of samples suspended in air. **c** Plot of steady-state temperature profiles around crystal growth front using parameters from global fit of data from **a** (green) and **b** (orange) to the model in Supplementary Information Section 4 shown for several velocities.

steep phase transition at a lower critical solution temperature (LCST) around 32 °C, leading to a large contraction when the temperature exceeds the LCST. Contraction rates in such gels are often limited by the formation of an impermeable polymer-rich skin layer at the gel–solution interface[3]. In order to ensure a fast response, we subjected the gel to freeze–thaw cycles during preparation, leading to the formation of macroscopic pores that aid in water transport[51]. As crystals grew in the underlying channel, confocal Z-stacks showed a contraction wave in the thermally responsive gel following the underlying crystal growth front (Fig. 5, Supplementary Movies 6 and 7); the thickness of the gel shrank to 80 μm within one minute before recovering slowly. Such wavelike contractions, which our group has previously explored using chemical triggers[13], recall peristaltic motions. Applying wavelike temperature profiles to monolithic thermo-responsive films, which can be purchased as coatings for cell culture dishes for the purpose of controllably detaching cells[52,53], accordingly converts a homogeneous deswelling response to a directional motion with a prescribed orientation and velocity. Nistor et al. previously engineered peristaltic actuation in thermoresponsive hydrogels using ohmic heating with complicated arrangements of over 10 heating elements that were activated in sequence[11]. The approach shown here achieves a similar effect with a significantly simpler setup; it may be further developed to move cargo such as settled particles in a desired direction or along a prescribed pathway.

We have engineered two further demonstrations to show that thermoresponsive actuation can be confined to masked regions (Fig. 6). First, a coverslip bearing a film of poly(N-isopropyla-crylamide) was again placed onto a channel containing a photo-patterned metastable hydrogel. This channel was thinner (0.5 mm) and was situated on a metal plate in order to facilitate heat dissipation from slow-moving crystal growth fronts; a large

region including masked and unmasked areas was observed under a microscope. Upon initiation of crystal growth, the thermally responsive gel contracted only over the masked regions of the underlying channel (Fig. 6a, b, Supplementary Movie 8). Next, we overlaid a sheet of tissue paper bearing a thin layer of dyed icosane wax (melting point = 35–37 °C) onto a channel containing a metastable hydrogel photo-masked with the word "HOT". As crystals grew within the gel, icosane selectively melted directly over the masked areas where crystallization proceeded quickly. Upon melting, the icosane soaked through the paper to reveal the underlying pattern (Fig. 6c, d, Supplementary Movie 9). In this system, transitions that occur at specific temperatures act in a similar way to the thresholding algorithm that produced Fig. 4b, d, only proceeding in areas where the temperature is allowed to exceed the transition temperature. When applied to the cell culture dishes mentioned above, this may enable the fabrication of patterned cell sheets via the selective liftoff of chosen areas; the hydrogels described in this work could quite conceivably be included as an enclosed active layer in disposable Petri dishes, for example.

The photopolymerization-based strategy we have presented here enables us to exert control over numerous aspects of the solidification process: the rates at which crystals grow in different domains of the material, the shape and directionality of the growth fronts, the size and geometry of the grains in the final crystalline material, and the peak temperature reached in each region. We have highlighted the potential utility of this strategy as a means of patterning thermally activated processes, but it may be relevant in other contexts, as well. Our results confirm that polymer additives provide a means of controlling the rate of heat release by solidifying phase change materials; this insight may also be useful in the design of systems where such materials are applied in bulk, such as in energy storage and thermal

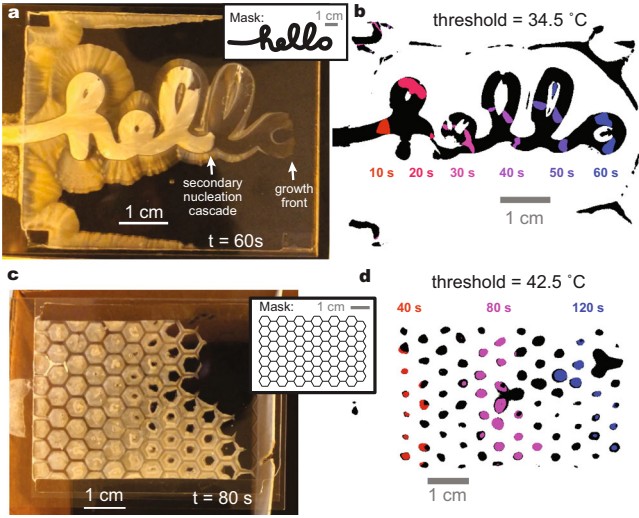

**Fig. 4 Spatiotemporal evolution of thermal profiles from patterned sodium acetate trihydrate crystal growth.** Temperature was monitored with an infrared camera. Precursor solution composition for all samples: 7.0 M sodium acetate, 2.8 M acrylamide, 13 mM N,N′-methylenebisacrylamide, 2 mM α-ketoglutaric acid in water; 0.5 mm spacer. **a** Image of crystal growth in a sample that had been photopolymerized through a mask spelling out the word "hello" with light blocked within the letters. Sample was crystallized on a cold plate held at 18 °C. Full optical and thermal video available (Supplementary Movie 4). **b** Sum of thresholded thermal images of the sample shown in (**a**). Regions that reached a threshold temperature of 34.5 °C over the course of sample crystallization are shown in black. Regions that exceeded this threshold at 10-s interval snapshots are shown in a spectrum from red to blue. **c** Image of crystal growth in a sample that had been photopolymerized through a hexagonal grid mask with light blocked by the lines. Sample was crystallized in air rather than on a heat sink; as such, heat was allowed to accumulate in areas where fronts converged. Full optical and thermal video available (Supplementary Movie 5). **d** Sum of thresholded thermal images of the sample shown in (**c**). Regions that reached a threshold temperature of 42.5 °C over the course of sample crystallization are shown in black. Regions that exceeded this threshold at 40-s interval snapshots are shown in red, magenta, and blue.

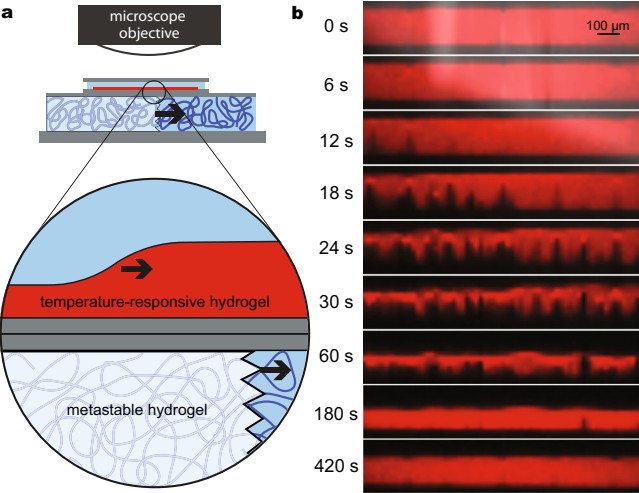

**Fig. 5 Coupling of a temperature wave in crystallizing gel to a contraction wave in an adjacent thermoresponsive hydrogel film. a** Schematic of experiment. A confocal microscope captures Z-stacks of a film of hydrated poly(N-isopropylacrylamide) bearing Nile blue acrylamide dye as sodium acetate crystallizes in a metastable hydrogel inside an underlying crystallization channel with a thickness of 2 mm. **b** Cross-sections of thermoresponsive gel film at various timepoints show a contraction wave. Red signal shows Nile blue acrylamide fluorescence ($\lambda_{ex} = 488$ nm, $\lambda_{em} > 492$ nm); white signal shows transmitted light. The transmitted light signal goes dark when crystals pass under the field of view; the contraction wave trails crystal growth by ~700 μm. Video showing XY, XZ, and YZ cross-sections available as Supplementary Movie 6; a similar experiment with a wider field of view and lower magnification is shown in Supplementary Movie 7.

management in buildings. It may also be possible to couple the patterned heating elements described in this work to a thermally conductive plate to spatially homogenize the heating response. The scheme would then function mainly as a means of patterning heating in time, as significantly more heat flux occurs when the crystal growth front passes through large unpolymerized areas compared to polymerized domains. Producing thermal pulses in this way using phase change materials that reach higher temperatures may enable the off-grid production of controlled thermal cycles such as those required for polymerase chain reactions. This work may additionally inform the design of processes for composite material fabrication, as we have been able to pattern boundaries between domains containing crystals with sizes and aspect ratios that differ by orders of magnitude (Supplementary Figs. 1–3, Supplementary Table 2), which will impart the domains with vastly different properties including stiffness, toughness, light scattering, and anisotropy.

Further investigation into the mechanism by which the polymerization state of a solution defines the kinetics of crystal growth is warranted, as are investigations into the applicability of this technique to different material systems. In the context of heat generation, choosing crystallizing species such as other salt hydrates[20] with different melting points, enthalpies of fusion, and crystal growth rates will define the shape and velocity of the temperature profiles produced, allowing further control over the thermal evolution of the material; a preliminary investigation of three other salt hydrates can be found in Supplementary Table 7. Additionally, although this manuscript has focused on developing and characterizing a single-use heat wave generation scheme, investigations into the reversibility of these systems are warranted. Further, other types of polymers and gels, including physical gels held together with supramolecular or other noncovalent interactions, may inhibit or otherwise modify the kinetics of crystal growth from metastable solutions in interesting ways. Finally, since the scheme presented here relies on polymers to control the rate of an exothermic process, using thermally responsive polymers in metastable solutions may present an opportunity to engineer feedback into crystallizing systems.

## Methods

**Precursor solution preparation.** All chemicals were purchased from Millipore Sigma except Nile blue acrylamide, which was purchased from Polysciences, Inc. Deionized water (Milli-Q), acrylamide, N,N′-methylenebisacrylamide, and anhydrous sodium acetate were combined in the desired proportions (see Supplementary Information Section 1) in a glass vial, sealed with a vent needle, loaded into an orbital incubator (Gyromax 727, Amerex Instruments), shaken at ~200 rpm at 65 °C until fully dissolved (~2 h), divided into aliquots, and cooled to room temperature. (Note: from this point forward, the solution is susceptible to sudden nucleation by errant seeds; the reader is advised to take care in handling the solution and thoroughly clean all surfaces it will touch in order to avoid unwanted crystallization. Dividing the solution into aliquots reduces the impact of such an event, or of any instances of spontaneous homogeneous nucleation, which is possible but infrequent and discussed in Supplementary Information Section 6.)

**Channel assembly.** 1 mm-thick glass slides (26 × 46 mm slides: Thermo Scientific #2646-001; 2 × 3 inch slides: VWR #48382-180) and No. 1 glass coverslips (24 × 60 mm slips: VWR #48393-106; 48 × 60 mm slips: Thermo Scientific Gold Seal Cover Glass) were rinsed with water and isopropyl alcohol, blown dry, plasma-activated in an $O_2$ atmosphere (Diener Femto PCCE, 5 min at 50% power),

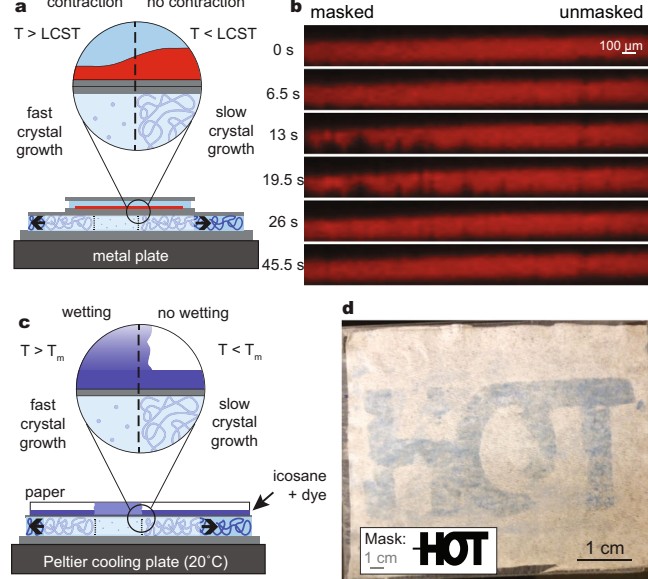

**Fig. 6 Coupling of exothermic crystal growth in patterned hydrogels to downstream phase changes that selectively occur in masked areas. a** Schematic of experiment similar to Fig. 5 in which a thermoresponsive hydrogel film was placed on a 0.5 mm-thick crystallization channel resting on a metal plate to promote heat dissipation. Fast crystal growth occurs in masked regions out of the plane of the page in this diagram; subsequent slow crystal growth outward through polymerized regions is shown with arrows. Contraction occurs selectively over masked regions. **b** Cross-sections of thermoresponsive gel film at various timepoints show selective contraction on masked side of film. Red signal shows Nile blue acrylamide fluorescence ($\lambda_{ex}$ = 488 nm, $\lambda_{em}$ > 492 nm). Video showing *XY*, *XZ*, and *YZ* cross-sections available as Supplementary Movie 8. **c** Schematic of experiment in which heat from crystallization in masked regions was used to melt a thin layer of dyed icosane wax (melting point = 35–37 °C), which then wet the paper above it to yield a visibly colored region. **d** Image of paper resulting from process outlined in (**c**) in which the hydrogel was photopolymerized through a mask containing the word "HOT." Full video available (Supplementary Movie 9).

submerged in a 10 vol% solution of (3-trimethoxysilyl)propyl methacrylate in dichloromethane for 1.5 h, rinsed twice in acetone, and blown dry again. Flat channels were assembled by sandwiching a spacer (3M VHB 4905 tape for all applications other than cross-polarization microscopy, where 3M double-sided Scotch tape was used) between a slide and a coverslip. Channels were rinsed with deionized water and blown dry immediately before filling with precursor solution as a cleaning measure to avoid unintentional nucleation.

**Photomask preparation**. Photomask designs were prepared as vector images, then cut into black vinyl adhesive decal or window cling material (Cricut) using a Cricut Maker. Masks were attached directly on the coverslip side of the channels, taking care to avoid trapping air bubbles between the mask and the coverslip.

**Metastable hydrogel synthesis**. 0.1 vol% of a stock solution of 2 M α-ketoglutaric acid in water was added to an aliquot of precursor solution, followed by brief vortex mixing. This solution was pipetted into a channel, which was placed under a UV lamp (UVP Blak-Ray B-100AP) at a distance of 3 cm on top of a matte black surface for 10 min.

**Preparation of thermally responsive hydrogel films**. A channel was prepared as above using an unmodified glass slide, a coverslip that had been functionalized with (3-trimethoxysilyl)propyl methacrylate, and Scotch tape as a spacer. A solution of 2.6 M *N*-isopropylacrylamide, 13 mM polyethylene glycol diacrylate ($M_n$ = 575), 2.7 mM Nile blue acrylamide, and 40 mM azobisisobutyronitrile in DMSO was prepared and injected into the channel. The sample was baked at 70 °C for 80 min and soaked in deionized water overnight. The coverslip, which at this point was covalently attached to the polymer network, was carefully peeled off of the glass slide, re-submerged in deionized water, subjected to two cycles of freezing in a freezer at 20 °C for ≥3 h and thawing in an oven at 55 °C, and stored underwater at

room temperature. Before imaging, the glass side of the coated coverslip was placed directly onto the coverslip side of a channel that had previously been filled with metastable precursor solution and photocured; the sample was subsequently observed under a confocal microscope during crystallization as described below.

**Preparation of thermal paper**. 1% (w/v) Oil Blue N dye was added to icosane and mixed at 65 °C in an orbital incubator until dissolved. The resulting solution was cast into a small Petri dish while still hot, then allowed to cool and solidify overnight. The resulting blue puck of icosane was rubbed onto a piece of tissue paper (Kimwipes, Kimberly-Clark) in the manner of a crayon until the paper was covered in colored wax. The tissue was then stuck, wax side down, to the coverslip side of a channel that had previously been filled and photocured through a mask using thermally conductive glue (Halnziye). The sample was crystallized on a cold plate as described below.

**Crystallization and imaging**. Samples were placed either on a Peltier cold plate (TC-48-20 Thermoelectric Temperature Controller, TE Technology, Inc.) or on a paper grid to suspend the sample in air depending on the desired extent of heat dissipation, coverslip side up. Crystal growth was nucleated using a needle bearing seed crystals of sodium acetate trihydrate. Growth was monitored using traditional and/or infrared (FLIR SC5000) cameras. Infrared images were collected using Altair (FLIR) and processed using purpose-written code in MATLAB (MathWorks). Crystal growth front velocities in Fig. 2 were obtained by processing traditional video in Tracker (Open Source Physics, https://physlets.org/tracker/). Cross-polarization videos were collected using a Zeiss AxioImager 2 microscope in transmitted light mode through perpendicular linear polarizers. Scanning electron micrographs were obtained using a JEOL JSM-6390LV microscope using accelerating voltages from 5 to 20 kV; samples were sectioned with a razor blade and sputter-coated with gold for 1 min each using a Denton Desk V sputtering system at 18 mA. Confocal microscopy Z-stacks were obtained using a Zeiss LSM 700 microscope at an excitation wavelength of 488 nm; emitted light above 492 nm was collected. When samples were placed on a glass stage, transmitted light was also collected; on a metal stage, it was not. Crystal growth experiments were performed in ambient conditions without monitoring humidity.

## Data availability
The growth rate, temperature, and crystal size data generated in this study are provided in Supplementary Data 1.

## Code availability
Source code used in Supplementary Section 3C is provided with this paper in Supplementary Software 1.

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

## Acknowledgements
This material is based upon work supported by the U.S. Army Research Office, Department of the Army, under grant number W911NF-17-1-0351 (J.A.), by the US Department of Energy (DOE), Office of Science, Basic Energy Sciences (BES) under award number DE-SC0005247 (J.A.), and by a Postdoc.Mobility fellowship from the Swiss National Science Foundation, grant number P400P2_180743 (T.B.H.S.). The authors thank Amos Meeks, Yanhao Yu, Hang Zhang, Haritosh Patel, and Michael Aizenberg for fruitful scientific discussions.

## Author contributions
T.B.H.S. and J.A. conceived the project and wrote the manuscript. T.B.H.S. collected all data and performed all analysis, modeling, and simulations.

## Competing interests
The authors declare no competing interests.
