## [Peer Review File · Nature Communications]

REVIEWER COMMENTS

Reviewer #1 (Remarks to the Author):

The authors have improved their manuscript based on reviewer feedback and have addressed several of the points we raised in our previous review, including the possibility of nucleation, the reasoning behind using sodium acetate, and the poor match of the measured pore sizes with their theoretically expected behavior. We reiterate that this work presents an interesting phenomenon of thermal pattern generation with a high degree of control and that we consider that the study was carried out rigorously and with strong theoretical support.

However, we consider that our main points have not yet been adequately addressed. As we commented in our previous review, there has been prior work on crystallization in metastable hydrogels, which the authors reference, and which has even used the same crystallization-dissolution reaction with sodium acetate as the authors use in this work. We consider that the main novelty of this work, namely the addition of patterning to this reaction and the subsequent use of these patterns to generate thermal patterns, has not been justified sufficiently and its impact is still unclear.

Our specific comments are:

1) The authors justify their study with a wide range of applications for thermo-responsive materials in Lines 28-31 (References 1-14) and later in the context of spatially patterned materials in Lines 39-40, where the authors reference previous work from their group. However, there is very little discussion on how their approach, one that stores the pattern and energy for the thermo-responsive behavior, would have any benefits in these applications.

The authors have added a discussion of a possible application for selective lift-off in cell culture dishes, but it is still unclear what and how other applications, from the many used to motivate this work, can benefit from this. We consider that this is a crucial point that needs to be addressed in detail in order to convey the impact of this work and justify its novelty.

2) We consider that the authors are not discussing in detail the differences between their proposed concept with other previous approaches of thermal pattern generation, like ohmic or photothermal approaches. In Lines 36-37 the authors state that “these methods require a connection to an external energy source at the moment of use, among other limitations”

What are those other limitations that the authors see?

What about the drawbacks of the authors' method? From the manuscript, it seems like the authors envision their device as a one-use device, which is not the case with other approaches. It also seems like there is an intrinsic volume limitation of their device to prevent crystallization. How can these limit the use of their proposed device?

The authors need to address these points in detail and by considering possible applications to convince that their work is a significant research improvement.

3) The authors have done a good job justifying their use of sodium acetate. However, it is still unclear if this concept is compatible with other salts. In particular, the authors have added graphs on the maximum temperature that can be reached as a function of the different relevant parameters, and they all show a temperature limit of ~50-60 °C. If a higher temperature was needed and a different salt was used, would this be possible with their fabrication approach?

Would other salts interfere with the polymerization of the unmasked areas?

Would the differences in crystallization speed persist?

Reviewer #2 (Remarks to the Author):

The authors described a strategy to adjust the fast, exothermic crystallization of sodium acetate from a metastable aqueous solution into trihydrate crystals within a polyacrylamide hydrogel. The polymerizable additives inhibit crystal growth to a significantly greater degree, which enables the patterning of pathways for rapid crystal growth by employing UV-initiated polymerization through photomask. The temperature difference and propagating heat waves were employed for the detailed analyses. In a word, this work discussed the crystal growth kinetics and heat evolution in hydrogel system in detail, which can meet the ultrahigh standards of Nature communications.

1) I want to know what happen if you change the tough polymer hydrogel to weak supramolecular hydrogel.

2) If you decrease the crosslinking density of polymer hydrogel, what happen.

Reviewer #1 (Remarks to the Author):

The authors have improved their manuscript based on reviewer feedback and have addressed several of the points we raised in our previous review, including the possibility of nucleation, the reasoning behind using sodium acetate, and the poor match of the measured pore sizes with their theoretically expected behavior.

We thank the reviewer for appreciating the improvements we have made in response to their previous review.

We reiterate that this work presents an interesting phenomenon of thermal pattern generation with a high degree of control and that we consider that the study was carried out rigorously and with strong theoretical support.

We appreciate the reviewer's approval of the rigor and theory development involved in this project.

However, we consider that our main points have not yet been adequately addressed. As we commented in our previous review, there has been prior work on crystallization in metastable hydrogels, which the authors reference, and which has even used the same crystallization-dissolution reaction with sodium acetate as the authors use in this work. We consider that the main novelty of this work, namely the addition of patterning to this reaction and the subsequent use of these patterns to generate thermal patterns, has not been justified sufficiently and its impact is still unclear.

Our specific comments are:

1) The authors justify their study with a wide range of applications for thermo-responsive materials in Lines 28-31 (References 1-14) and later in the context of spatially patterned materials in Lines 39-40, where the authors reference previous work from their group. However, there is very little discussion on how their approach, one that stores the pattern and energy for the thermo-responsive behavior, would have any benefits in these applications.

In response to this comment, we have discussed further the benefits of our approach in the applications listed and have made several changes to the text of the manuscript:

- We have removed several of the less relevant applications of thermoresponsive materials we mentioned at the beginning of the introduction (lines 28-30) – these had been intended to provide context about the current use of thermoresponsive materials in general rather than to imply that our patterned heat source is at the stage of being ready to be applied in each scheme. We have removed the most obviously extraneous examples.
- We added a portion at the end of the first paragraph of the introduction (lines 45-52) to discuss how a patterned thermal energy storage system such as ours might be useful in applications involving dynamic topography.

The altered sections of the manuscript have been reproduced below in green with the changed portions highlighted in yellow:

First paragraph of introduction (lines 26-52 are reproduced here):

Thermally responsive materials such as hydrogels, liquid crystals, shape-memory alloys, and waxes display temperature-dependent behaviors that can transduce heat into optical, mechanical, or chemical changes. Such materials have found wide applications, including in cell culture and tissue engineering,^{1,2} antifouling surfaces,³ microfluidics,^{4,5} soft robotics,^{6,7} and as a component of complex cascades.^{8,9}

First paragraph of introduction (lines 41-52):

The development of materials with dynamic surface topography is a focus of our group; we have applied such surfaces to antifouling applications,¹² the movement of cargo,³ the control of cultured cells,² and the actuation of rigid structures.¹³ We are therefore interested in developing heat sources that store both the energy and the pattern required to induce complex frontal behaviors such as patterned waves in thermally responsive materials on demand. For example, a scheme using dynamic swelling or contraction waves to move cargo on a gel film toward a destination requires the swelling or contraction to be directional;¹⁴ such directionality must be programmed. Likewise, rigid gel-embedded structures bend in a direction defined by the gradients involved in the gel's contraction;¹⁵ controlling the direction of actuation requires programming the gradients. Pre-defining such patterns using a scheme that also stores the energy required to operate may enable such dynamic surfaces to be incorporated into devices that can be used “in the field.”

The authors have added a discussion of a possible application for selective lift-off in cell culture dishes, but it is still unclear what and how other applications, from the many used to motivate this work, can benefit from this. We consider that this is a crucial point that needs to be addressed in detail in order to convey the impact of this work and justify its novelty.

We thank the reviewer for this comment. We believe the changes in response to the comment above partially address this by making the value that patterned heating can add to dynamic topography schemes apparent up-front. However, we believe the reviewer is also asking for some more specific visions of applications of the phenomenon we describe; accordingly, we have made several further changes to the text, outlined below.

- We have added text in the Results & Discussion section (lines 307-308) discussing the possibility of using this work as a soft robotic scheme to move cargo peristaltically.
- To further motivate the work, we have made an addition to the bottom of the second paragraph (lines 68-72) in which we suggest the utility of controlling the time-course of heat release in macroscopic metastable phase change materials. We then follow up on this possibility near the end of the Results & Discussion section (lines 353-359) by suggesting that pulsed heat release over time would be useful in the context of controlled thermal cycling for (for example) polymerase chain reactions “in the field.”

The following changes have been made to the manuscript:

Second paragraph of introduction (lines 62-72):

Supercooled phase change materials have found applications as triggerable heat sources in the context of portable hand heating,²¹ cold-start automotive engine heating,²² building-scale air and water heating,²³ and long-term solar energy storage.²⁴ These applications are mostly intended to raise temperatures in a bulk volume for human comfort or improved device function; accordingly, the spatial and temporal evolution of the thermal profiles produced by triggerable phase change materials have rarely been engineered beyond the shape of the reservoir. However, as the solidification of metastable phase change materials generally proceeds from a defined nucleation point, the ability to control the velocity of the crystallization front throughout a reservoir may be useful, enabling control over the time-course of heat release to produce delays or pulses as desired.

Tenth paragraph of Results & Discussion section (lines 303-307):

Nistor et al. previously engineered peristaltic actuation in thermoresponsive hydrogels using ohmic heating with complicated arrangements of over 10 heating elements that were activated in sequence.¹¹ The approach shown here achieves a similar effect with a significantly simpler setup; we believe that it may be further developed to move cargo such as settled particles in a desired direction or along a prescribed pathway.³

Second-to-last paragraph of Results & Discussion section (lines 347-358):

We have highlighted the potential utility of this strategy as a means of patterning thermally activated processes, but it may be relevant in other contexts, as well. Our results confirm that polymer additives provide a means of controlling the rate of heat release by solidifying phase change materials; this insight may also be useful in the design of systems where such materials are applied in bulk, such as in energy storage and thermal management in buildings. It may also be possible to couple the patterned heating elements described in this work to a thermally conductive plate to spatially homogenize the heating response. The scheme would then function mainly as a means of patterning heating in time, as significantly more heat flux occurs when the crystal growth front passes through large unpolymerized areas compared to polymerized domains. Producing thermal pulses in this way using phase change materials that reach higher temperatures may enable the off-grid production of controlled thermal cycles such as those required for polymerase chain reactions.

2) We consider that the authors are not discussing in detail the differences between their proposed concept with other previous approaches of thermal pattern generation, like ohmic or photothermal approaches. In Lines 36-37 the authors state that “these methods require a connection to an external energy source at the moment of use, among other limitations” What are those other limitations that the authors see?

In response to this comment, we have expanded upon the description of the limitations of ohmic and photothermal approaches in the manuscript; the relevant portion is reproduced below.

First paragraph of introduction (lines 30-41):

Most deployments of thermally responsive materials rely on bulk changes in temperature to produce a spatially homogeneous response; some produce more complex responses by judicious arrangement of the responsive domains of the material.⁶ Coupling such materials to an upstream heat source that dynamically produces thermal patterns with spatial resolution is another way to impart complexity to their responses. This is frequently accomplished using photothermal^{2,10} or ohmic¹¹ heating. However, these methods have limitations: photothermal heating requires transparent samples, a suitably high-powered light source and a means of patterning light and moving this pattern with sufficient resolution at the point of use, while achieving patterned ohmic heating requires a connection to external power and complex electrical circuitry – individually addressable resistive heating elements must be distributed throughout the area of interest, and a control scheme must be used to activate them sequentially.

What about the drawbacks of the authors' method? From the manuscript, it seems like the authors envision their device as a one-use device, which is not the case with other approaches. It also seems like there is an intrinsic volume limitation of their device to prevent crystallization. How can these limit the use of their proposed device? The authors need to address these points in detail and by considering possible applications to convince that their work is a significant research improvement.

We agree that we should more thoroughly discuss the possible drawbacks of our method including reversibility and spontaneous nucleation. We stress that we view these as *possible* drawbacks, as further development may enable multiple cycles of patterned freezing and melting and further characterization may confirm that spontaneous nucleation presents negligible risk (depending on the context). In response to this comment, we have made two changes to the text of the manuscript:

- We have added a sentence to the concluding paragraph of the manuscript (lines 369-371) to raise the possibility of reversibility while acknowledging that thus far, we have characterized the scheme for use in single-use applications.
- We have added a passage to Section S7 that considers the size limitations imposed by the possibility of spontaneous nucleation. We point to the example of hand warmers as an example where 2-3 orders of magnitude larger volumes of supersaturated sodium acetate than we study have been specifically sold for use at low environmental temperatures and to be carried around roughly outdoors. (Anecdotally, we once purchased a box of these for demonstrations; spontaneous nucleation was not a noticeable problem over a period of weeks). It is on this basis that we can justify our optimism that even deployments of substantially larger volumes of these solutions than we have studied may be useful.

The following changes have been made to the manuscript:

Final paragraph of Results & Discussion section (lines 369-371):

Additionally, although this manuscript has focused on developing and characterizing a single-use heat wave generation scheme, investigations into the reversibility of these systems are warranted.

Section S7 (SI page 17):

However, for the solutions and volumes described in this work, the timescales involved are long enough that spontaneous nucleation is observed infrequently; anecdotally, we have vials of uncrystallized metastable precursor solutions that have been lying around the lab at or below room temperature for over a year, an observation corroborated by Sandnes and Rekestad.²⁴ That said, it does happen -- unperturbed precursor solution vials are occasionally observed to crystallize on a timescale of days or weeks; accordingly, we recommend dividing precursor solutions into aliquots to hedge against this possibility. Deployments of metastable solutions in large contiguous quantities may be limited by primary homogeneous nucleation. However, commercially available hand warmers containing over 100 mL of supersaturated sodium acetate solution (2-3 orders of magnitude more than used in this work) resist nucleation well enough to be useful at winter temperatures while being carried outdoors, indicating that even quite substantial amounts of solution can be stable enough for applications where the cost of accidental crystallization is minimal.

3) The authors have done a good job justifying their use of sodium acetate. However, it is still unclear if this concept is compatible with other salts. In particular, the authors have added graphs on the maximum temperature that can be reached as a function of the different relevant parameters, and they all show a temperature limit of ~50-60 °C. If a higher temperature was needed and a different salt was used, would this be possible with their fabrication approach? Would other salts interfere with the polymerization of the unmasked areas? Would the differences in crystallization speed persist?

The question of compatibility with other salts is certainly an important one. To address it, we have collected new velocity data on acrylamide solutions containing several other salt hydrates known to be metastably supercoolable, resulting in the new Table S7 (shown below), which clearly shows large differences in crystallization speed between the polymerized and unpolymerized states for each case. In particular, we note that the reviewer correctly predicted that the salt chosen could affect the polymerization process, as using sodium thiosulfate led to a visible decrease in gel strength (though polymerization still led to a marked reduction in crystal growth rate).

On the question of temperature: it would be theoretically possible to choose a different salt with a higher melting point and enthalpy of fusion and reach temperatures that exceed 50-60°C; the dotted red curve in the middle plot in Figure S9 (where β is varied) peaks at 79°C, for example.

However, there are some issues that make using these different salts less straightforward than sodium acetate:

- First, sodium acetate is able to reach its peak temperatures of $\sim 45^{\circ}\text{C}$ because its crystal growth rate through unpolymerized media is uncommonly fast, in the mm/s range (compared to the 60-120 $\mu\text{m/s}$ range as in the formulations shown above). As discussed throughout the manuscript, the peak temperature of the wave is strongly dependent on its velocity (see Figs. 3 and S8); slower crystal growth would produce heat waves with markedly lower peak temperatures. A fast-crystallizing formulation would therefore be critical. We have added “crystallizes quickly” to sodium acetate’s list of positive qualities on line 113 of the manuscript.
- Second, as is briefly discussed in Section S7, the probability of spontaneous nucleation is correlated with the degree of supercooling of the salt. While sodium acetate solutions have proven to be quite resistant to unwanted nucleation, the higher-melting salts required to produce high peak temperatures would be more highly supercooled at room temperature and thus more prone to it – indeed, we note in Table S7 that we encountered this when looking at magnesium nitrate solutions that had been supercooled by nearly 70°C . A nucleation-resistant formulation would thus be important to achieve high peak temperatures, as well.

The new Table S7 is reproduced below. The variable concentrations of acrylamide shown in this table are due to the fact that each salt has a different solubility and can be co-solvated with acrylamide to a different degree, which must be determined by trial and error for each compound.

Table S7: Growth rate of other salt hydrate crystals in unpolymerized and polymerized acrylamide solutions. Polymerization was achieved via UV irradiation for 10 minutes using 20 mM α -ketoglutaric acid. Average rates \pm s.e.m. shown (where applicable); $N = 2$ or 3.

Salt hydrate	Melting point ($^{\circ}\text{C}$)	[salt] (M)	[acrylamide] (M)	Crystal growth rate through unpolymerized solution ($\mu\text{m/s}$)	Crystal growth rate through polymerized gel ($\mu\text{m/s}$)	Notes
$\text{Mg}(\text{NO}_3)_2 \cdot 6\text{H}_2\text{O}$	89	3.7	2.6	91 ± 0.3	2.6 ± 0.1	Nucleation in polymerized case proceeded spontaneously from locations in channel interior following cooling to 20°C on Peltier plate.
$\text{Na}_2\text{S}_2\text{O}_3 \cdot 5\text{H}_2\text{O}$	48	3.5	1.9	120	5.9 ± 2.2	Gelation was noticeably inhibited.
$\text{Ca}(\text{NO}_3)_2 \cdot 4\text{H}_2\text{O}$	43	5.7	3.0	59 ± 3	No visible growth after 6 hours	

Reviewer #2 (Remarks to the Author):

The authors described a strategy to adjust the fast, exothermic crystallization of sodium acetate from a metastable aqueous solution into trihydrate crystals within a polyacrylamide hydrogel.

The polymerizable additives inhibit crystal growth to a significantly greater degree, which enables the patterning of pathways for rapid crystal growth by employing UV-initiated polymerization through photomask. The temperature difference and propagating heat waves were employed for the detailed analyses. In a word, this work discussed the crystal growth kinetics and heat evolution in hydrogel system in detail, which can meet the ultrahigh standards of Nature communications.

We are very grateful to the reviewer for their positive appraisal of our work.

1) I want to know what happen if you change the tough polymer hydrogel to weak supramolecular hydrogel.

We are also quite curious about this; however, the assembly of supramolecular or “physical” hydrogels in supersaturated salt solutions is non-straightforward. The problem is this: the solutions that we examine in this work have an ionic strength of 7 M, and the salt we use is kosmotropic, meaning that this solution is extremely prone to “salting out” macromolecules. We have found that it is basically impossible to dissolve common macromolecular gel components such as gelatin, agarose, or poly(vinyl alcohol) in solutions with ionic strengths this high; the only thing that has reliably worked so far has been to form macromolecular networks within the metastable solutions from small, soluble molecules (i.e. by polymerizing acrylic monomers). Further, we would expect that supramolecularly-linked gels held together by the interaction of charged groups, such as polyelectrolyte complexes, would not cohere at such high ionic strengths due to charge screening (see Wang and Schlenoff, *Macromolecules* 2014, 47, 3108-3116).

In short, we are actively investigating different gel types that can be used with this system, but the extreme salt concentration limits this repertoire considerably, and therefore we have left this investigation outside the scope of this first report.

To address this comment, the following sentence was added to the final paragraph of the manuscript (lines 371-374):

Further, other types of polymers and gels, including “physical” gels held together with supramolecular or other noncovalent interactions, may inhibit or otherwise modify the kinetics of crystal growth from metastable solutions in interesting ways.

2) If you decrease the crosslinking density of polymer hydrogel, what happen.

We were surprised to find that the crosslinking density of our hydrogels has relatively little bearing on the crystal growth front velocity. This is shown in Figure 2 B, reproduced below. This lack of a dependance on the cross-linking density informed our choice of possible inhibition mechanisms to discuss in Section S4, which are mostly agnostic to crosslinking (with the exception of the Gibbs-Thomson/pore penetration mechanism, which fits our data poorly and therefore is not likely to be operating, as we conclude in the text).

To address this comment, we further emphasized the independence of crystal growth velocity from the polymer's crosslinking density by strengthening the language where this figure is introduced in the Results section of the manuscript:

Third paragraph of the Results and Discussion section (lines 161-163):

Interestingly, the crystal growth rate in polymerized solutions was not significantly changed by the addition of a cross-linker at concentrations in the range of 0 - 20 mM (Figure 2B); however, we included one in photomasking experiments in order to prevent the loss of the pattern via diffusion over time.

Figure 2. Comparison of linear growth rates of sodium acetate trihydrate crystals in samples containing 7.0 M sodium acetate and various polymerized and unpolymerized additive compositions. $N = 3$ for all points except 2.8 M acrylamide without cross-linker in both graphs, where $N = 6$. Crystallizing samples were placed on a cold plate held at 20°C. **A.** Comparison of growth rates through polymerized and unpolymerized solutions containing different concentrations of acrylamide without a cross-linker. **B.** Comparison of growth rates through polymerized and unpolymerized solutions containing 2.8 M acrylamide and various concentrations of N,N' -methylenebisacrylamide cross-linker.

REVIEWERS' COMMENTS

Reviewer #1 (Remarks to the Author):

Response to author rebuttal regarding “Patterned crystal growth and heat wave generation in hydrogels” (NCOMMS-21-33858) by Thomas B. H. Schroeder, Joanna Aizenberg.

The authors have significantly improved their manuscript based on reviewer feedback and have provided additional experimental data and motivation for their work which has improved the quality of their work. We consider that the authors have adequately provided context for possible applications of their patterning approach and have expanded on the differences with previous approaches. We also consider that the additional data on different salts as well as the text added on the volume limitations correctly shows the possible drawbacks of their approach and as such provides useful information to the readers.

With the modifications made, we recommend the publication of the revised version of the manuscript to Nature Communications.

Reviewer #2 (Remarks to the Author):

We are satisfied with the revision done by the authors.

Reviewer #1 (Remarks to the Author):

Response to author rebuttal regarding “Patterned crystal growth and heat wave generation in hydrogels” (NCOMMS-21-33858) by Thomas B. H. Schroeder, Joanna Aizenberg.

The authors have significantly improved their manuscript based on reviewer feedback and have provided additional experimental data and motivation for their work which has improved the quality of their work. We consider that the authors have adequately provided context for possible applications of their patterning approach and have expanded on the differences with previous approaches. We also consider that the additional data on different salts as well as the text added on the volume limitations correctly shows the possible drawbacks of their approach and as such provides useful information to the readers.

With the modifications made, we recommend the publication of the revised version of the manuscript to Nature Communications.

We are grateful to the reviewer for their approval, as well as for their insightful comments throughout the review process; their feedback helped us make this a better manuscript.

Reviewer #2 (Remarks to the Author):

We are satisfied with the revision done by the authors.

We are grateful to the reviewer for their approval, as well as for their insightful comments throughout the review process; their feedback helped us make this a better manuscript.